# Temporal Analysis of Climate Change Impact on the Spread and Prevalence of Vector-Borne Diseases in Campania (2018–2023)

**DOI:** 10.3390/microorganisms13020449

**Published:** 2025-02-18

**Authors:** Federica Gargano, Roberta Brunetti, Marialuisa Buonanno, Claudio De Martinis, Lorena Cardillo, Pellegrino Fenizia, Antonietta Anatriello, Giuseppe Rofrano, Luigi Jacopo D’Auria, Giovanna Fusco, Loredana Baldi, Esterina De Carlo, Maria Ottaiano

**Affiliations:** 1Department of Epidemiology and Biostatistics Regional Observatory (OREB), Istituto Zooprofilattico Sperimentale del Mezzogiorno, 80055 Naples, Italy; federica.gargano@izsmportici.it (F.G.); marialuisa.buonanno@izsmportici.it (M.B.); pellegrino.fenizia@izsmportici.it (P.F.); loredana.baldi@izsmportici.it (L.B.); maria.ottaiano@izsmportici.it (M.O.); 2Department of Animal Health, Istituto Zooprofilattico Sperimentale del Mezzogiorno, 80055 Naples, Italy; claudio.demartinis@izsmportici.it (C.D.M.); lorena.cardillo@izsmportici.it (L.C.); giovanna.fusco@izsmportici.it (G.F.); 3Department of Experimental Medicine, University of Campania “Luigi Vanvitelli”, 80138 Naples, Italy; antonietta.anatriello@unicampania.it; 4Regional Center of Pharmacovigilance and Pharmacoepidemiology of Campania Region, 80138 Naples, Italy; 5National Reference Center for the Analysis and Study of Correlation between Environment, Animal and Man, Istituto Zooprofilattico Sperimentale del Mezzogiorno, IZS Mezzogiorno, Via Salute 2, 80055 Naples, Italy; giuseppe.rofrano@izsmportici.it (G.R.); jacopoluigi.dauria@izsmportici.it (L.J.D.); 6National Reference Centre for Hygiene and Technologies of Water Buffalo Farming and Productions, Istituto Zooprofilattico Sperimentale del Mezzogiorno, 80055 Naples, Italy; esterina.decarlo@izsmportici.it

**Keywords:** climate change, vectore-borne-desease, regression model, temporal analysis

## Abstract

Vector-borne infections (Arbovirosis) represent a significant threat to public health worldwide. Climate change, currently a global problem, seems to contribute to the incidence and prevalence of autochthonous and imported cases of arbovirosis in Europe. The aim of this work is to evaluate the impact of climate change on the occurrence and spread of arbovirosis in order to offer concrete ideas to the competent authorities and modulate surveillance plans on the basis of risk assessment. The results of official controls carried out from 2018 to 2023 on animals subjected to the Surveillance Plans for Blue Tongue, West Nile and Usutu viruses in the Campania Region were analyzed. Statistical analyses were performed using R software (version 4.5.0). The possible correlation between the prevalence of infections and climate parameters was evaluated with the “cross-correlation time series” (*p*-value < 0.05). The generalized linear mixed model showed that for each unit increase in humidity and temperature, the probability of disease spread increased by 4.56% and 7.84%, respectively. The univariate logistic regression model and the odds ratio were necessary to evaluate the possible risk related to the proximity to wetlands or to bodies of water: in the past few years, these have represented a risk for the persistence and spread of arbovirosis in the Campania region.

## 1. Introduction

Vector-borne diseases constitute an important public health problem: the WHO estimates that every year they cause over a billion human cases and 1 million deaths, representing approximately 17% of total cases of communicable diseases [1]. There are over 100 viruses classified as arboviruses capable of causing human disease. Climate change has a direct impact on the distribution and activity of vectors, influencing factors such as temperature, precipitation and humidity, which are crucial for the life cycle of insects and their ability to transmit pathogens. Rising global temperatures, more extreme weather events and changing precipitation patterns are factors that are modifying the distribution of infectious diseases worldwide [2,3]. Regions at risk include not only the tropics, but also temperate and subtropical areas, which could see an expansion of vectors and a consequent increase in vector-borne disease cases. Recent studies have documented the emergence of previously geographically restricted diseases, with malaria and dengue-carrying mosquitoes now found in territories further north, such as Europe and the United States [4,5]. For example, the spread of dengue in areas such as the Mediterranean, which were not previously subject to this disease, represents clear evidence of the influence of global warming on vector-borne diseases [6]. Furthermore, the impact of climate change is not limited to the simple geographical expansion of vectors, but can also alter transmission patterns, seasonality and intensity of epidemics. Longer heat periods and heavy rainfall can extend the vector activity season, increasing the likelihood of infectious disease transmission. The situation is further complicated by globalization, which facilitates the spread of diseases through international travel and trade [7]. In this context, a global view of vector-borne diseases is essential, as climate change is creating new risks not only for tropical regions, but also for countries with temperate climates. An analysis of global trends is therefore essential to fully understand the implications of climate change on public health. A promising approach to address this challenge is the use of advanced technologies, such as Geographic Information Systems (GIS) and remote sensing, which allow monitoring and forecasting the distribution of vectors in relation to ecological and climatic variables. The integration of satellite data with environmental suitability models allows mapping areas at risk of disease transmission, providing crucial information for planning and implementing targeted control and prevention measures. The predictive approach, which exploits these technological tools, not only optimizes the management of health resources, but is part of the concept of “One Health”, which recognizes the connection between human, animal and environmental health. A concrete example of the application of these technologies is represented by the study by Ippoliti et al. (2024) [8], which uses a multivariate clustering method to define ecological regions in Italy and improve the surveillance of vector-borne diseases. This study demonstrates how predictive analytics can support health policies, improving the capacity to respond to emerging diseases and reducing risks to public health. In Italy, there are both native arboviruses, including West Nile disease, Usutu virus infection, Toscana virus infection and tick-borne viral encephalitis, and predominantly imported arboviruses, such as the infections caused from Chikungunya, Dengue and Zika viruses, as well as bluetongue (BTV) [9]. Usutu (USUV) and West Nile (WNV) viruses are mosquito-borne neurotropic flaviviruses belonging to the Flaviviridae family and the Flavivirus genus. The natural transmission cycle of WNV and USUV involves mosquitoes and birds, while mammals are believed to be accidental hosts. WNV is now the most widespread virus among the flaviviruses reported on almost all continents. In the last two decades, their presence has been confirmed in several European and Mediterranean countries, becoming endemic in some of them. USUV was originally isolated in South Africa and first detected in Europe in 2001, where it caused severe mortality in Austrian blackbird populations. In the following years, USUV spread to Central and Eastern Europe, affecting wild birds, including populations from Italy, where the virus has probably been circulating since 1996. The USUV was also first detected in the UK in 2020 [10]. Bluetongue is an arthropod-borne viral infection that is notifiable in several countries and causes significant economic losses and serious concerns for the ruminant trade. BTV is a non-contagious RNA virus spread by *Culicoides* bite, which has significantly invaded Europe in recent decades [11]. BTV was responsible for well-known epidemics in the late 1990s and early 2000s and is now considered an endemic and re-emerging virus throughout Europe. Due to the severe symptoms exhibited by infected animals, BTV infection causes economic losses and impacts international livestock trade (estimated at up to USD 3 billion) [12]. Although BTV primarily results in clinical symptoms in sheep, cattle serve as reservoir hosts and play an important role in the transmission and epidemiology of BTV because they exhibit prolonged viremia (rarely associated with reproductive failure and low milk production) [13]. Surveillance for this virus relies on identification of the pathogen in vectors and symptomatic hosts, as well as the detection of specific antibodies. There are numerous factors that influence the transmission and distribution of arboviruses, linked to the dynamics and interactions among pathogen, vector, host and environment. Climatic conditions have direct and indirect influences on the competence of the vector (the ability to acquire, maintain and transmit the virus), on the dynamics of the vector population and on the rate of replication of the virus within the mosquito. The dynamic evolution of ecosystems together with climate changes have played and still play an important role in the spread of these diseases, contributing to their spread in countries where these pathologies were not present and influencing their persistence in new areas. The impact of climatic factors (temperature, precipitation, relative humidity and winds) on the epidemiology of arboviruses, in fact, is increasing considerably as a function of the current continuous climate change, which influences the emergence of arboviruses such as malaria, dengue and WNV by altering its frequency, extent, distribution and seasonality. Many indeed, in fact, were once considered exotic in our country, have now become epidemic and/or sometimes endemic. The Intergovernmental Panel on Climate Change (IPCC) lists vector-borne diseases among the events most likely to change due to global warming. On the other hand, these diseases, particularly sensitive to climate fluctuations, could act as an alarm to focus attention on the threats of climate change [14]. In this regard, Governments are obliged to consider the effect of climate change in their prevention, control and surveillance perspectives in all countries of the world. To mitigate the impacts of CC on the transmission ecology of vector diseases, it is important to implement effective surveillance and control measures that cannot ignore the study of the effect of climate change on the onset and spread of these infections in order to modulate and intensify early detection in certain periods of the year, thus ensuring efficient and effective use of the resources available to protect health.

The One Health approach, based on the cooperation and interdisciplinarity of various professional figures, could be the only valid and truly effective alternative to deal with possible threats emerging from the human–animal-environment interface [15]. The organization of a territorial cooperation network constitutes a valid innovative reference model in the integration of the different activities and experiences of human medicine and veterinary medicine through the intensification and integration of animal surveillance and prevention plans with human and environmental (entomological). In recent decades, an increase in cases of vector-borne infections, both emerging and non-emerging, has been observed in many Italian regions [15]. In the Campania region, there has been an increase in WND cases in recent years versus no, or almost no, detection of infection in the years prior to 2022. The objective of this study is to determine whether climatic parameters, such as temperature and humidity, represented risk factors for the spread of Arbovirosis in the Campania region in the years 2018–2023. The results may be useful to the competent authorities to modulate existing surveillance plans.

## 2. Materials and Methods

An analytical observational case–control study was conducted on the animal population controlled for WNV, USUTU and BTV in the Campania Region in the period 2018–2023. In this study, the subjects (cases) in whom an event occurred (infected animals) and their history of exposure to possible risk factors are established retrospectively and compared with controls (negative animals).

The data sources used are SIGLA (laboratory management system of the Istituto Zooprofilattico Sperimentale del Mezzogiorno), Archive software of the Department of Agriculture of the Campania Region on climate data (accessed on 1 February 2024) (portal where daily measurements relating to various climate parameters, including Temperature and Humidity, detected by various municipalities in the Campania Region are recorded), ISPRA (accessed on 10 July 2024) (Higher Institute for Environmental Protection and Research, from which the Shp file of the wetlands and water bodies of the Campania Region were obtained) and Vetinfo/BDN (accessed on 3 April 2024) (the National Database, which represents the information system in which information on establishments, farms, animals, movements and slaughters is recorded according to the methods established for the different species).

The different BDN extractions (2018–2023) were queued in order to obtain a single file for each species. To associate the geographical coordinates (latitude, longitude) with each herd, the last year of registration in BDN was considered in order to avoid duplication if the coordinates had been changed, over the years, and entered correctly following notification by the Local Competent Authorities.

The software used for data collection, cleaning, processing and statistical analysis are Microsoft office package (Excel-Access), R-studio version 4.5.0 and QGIS 3.24.2.

The data provided by SIGLA were used to collect the results (positive/negative) of the serological and PCR tests, carried out on individual animals relating to the diseases examined (BTV, WNV, USUTU). The animal was considered infected if it had at least one positive result in the confirmatory tests carried out by the Reference Center for Exotic Diseases (CESME); similarly, the establishment was considered positive if it had at least one positive animal. The monthly and annual prevalence of infection was calculated, both at provincial and regional level, through the ratio between the total number of positive animals and the total number of animals tested. The distribution of companies by outcome, over the 5 years, for each province of the Campania Region was depicted on cartography.

The environmental parameters were downloaded from the Archive of the Department of Agriculture of the Campania Region in Excel format covering the entire analysis period.

The following statistical indices were calculated in order to obtain a single value per month and per province:Average temperature (min, max and average) of the air per month (where present);Average humidity (min, max and average) of the air per month (where present);Median of humidity and average air temperature per month;The number of days with temperature and humidity above average.

Historical series relating to both prevalence and temperature and humidity were constructed for the Campania Region in the period examined. Working with time series, it is common to have missing data (missing values) that need to be handled before performing further analysis. In our work, a linear interpolation was carried out, so that the time series were not free of missing values. Linear interpolation is a powerful and simple tool for estimating missing values or constructing new points within a known data range, based on the assumption that the change between data points is linear. The value is calculated linearly using the closest observed values on both sides of the missing value in the distribution. It is widely used in various fields, including time series analysis, to ensure complete and continuous data.

In order to evaluate a possible correlation between the prevalence of vectorial diseases and climate parameters, the “Cross Correlation time series” correlation test was implemented [16].

The Cross Correlation Function (CCF) measures the correlation between two time series at different lags (lag). CCF values can be positive or negative, and each sign has a specific meaning in terms of the relationship between the two time series, at each positive or negative lag [17].

Positive values of the CCF indicate a direct proportionality between the two time series under examination; meanwhile, negative CCF values indicate an inverse proportionality between the two time series under examination.

A negative lag means that the Y series (prevalence) follows the X series (environmental parameter); conversely, a positive lag means that the X series follows the Y series.

The cross-correlation was applied on the residuals in order to remove, through the factorization of the data, the seasonality present in the analyzed time series. A maximum delay of 7 months was used with the aim of introducing noise into the analysis [18]. 

A generalized mixed-effects regression model was constructed considering prevalence as outcome and environmental parameters as covariates using a Gamma distribution, since prevalence showed strong right-skewness. The mean of the mean temperature and the median of the mean humidity were taken into consideration as environmental indices since, from the cross-correlation function, they are the ones that showed a statistically significant association.

One of the fundamental assumptions for building multivariate regression models is the absence of multicollinearity. Multicollinearity is a phenomenon that occurs in regression analysis when two or more independent variables are highly correlated, making it difficult to distinguish the specific effect of each variable on the dependent variable. Before applying the statistical tests to verify the correlation between the variables, the normality of the data was checked using the Shapiro–Wilk test.

To verify the correlation between the two variables (T and U), the non-parametric Spearman’s rho test was applied. Univariate regression models were built; subsequently, a multivariate regression model was built given the lack of multicollinearity between the two covariates examined (T and U).

The models were built considering the values of the covariates, in which the CCF showed a direct proportionality with the time series of the prevalence of negative lags. Negative CFF values at negative lags and CFF values at positive lags were not considered, as it is not plausible that the prevalence happens before the increase or decrease in environmental parameters (T and U).

For temperature, the values at lag −2 (the strongest and most significant CCF) and then the temperature values up to lag −5 were considered to build the model. For humidity, humidity values up to a maximum phase shift of −2 were considered.

Furthermore, in order to evaluate any risks deriving from the location of the factories near bodies of water and/or wetlands, a univariate logistic regression model was built for each year.

The dataset used consists of 6949 observations. The dichotomous covariates (Yes/NO) referred to the location of the herd within 5 km from the boundaries of wetlands and/or bodies of water or located in a wetland were analyzed.

A radius of 5 km was chosen as it represents the maximum flight radius of the mosquito vectors of the infections considered [19]. For establishments and wild animals found dead without geographical coordinates, the center of corresponding municipality was used (less than 20% of reports out of the total).

The response variable in this case is dichotomous (presence/absence of infection in the company). Before building the aforementioned model, Fisher’s test was applied to evaluate any statistically significant association with the outcome of the study by year.

## 3. Results

Figure 1 shows the distribution of positivities in the period examined for each province of the Campania Region. Wetlands and bodies of water are also depicted in the cartography (Figure 1).

The descriptive analysis of the considered period shows an increase in prevalence starting from August, with peaks in the autumn (September–November). During the summer period, especially in July, there was a decline in prevalence, which is common to all years, followed by a rapid increase in autumn. Prevalence in 2023 peaked much higher than in previous years; this could suggest a greater impact of external factors, such as climate change (Figure 2).

The average temperature trend (Figure 3) highlights the climatic seasonality typical of Campania, with mild winters and hot summers. The trend in average temperatures follows a predictable profile, with lower temperatures in the winter months and the peak between July and August, especially for the year 2023. An increase in temperatures can favor the reproduction and activity of vectors. The peak observed in summer months could coincide with the phase in which the vectors are more active, increasing the risk of transmission; actually, vector-borne diseases tend to increase in warm seasons, in areas where temperature and humidity allow the survival and proliferation of vectors [11,12].

As regards the humidity graph (Figure 3), a predictable seasonal trend is highlighted in Campania, with higher values in winter and autumn and lower values in spring and summer. A significant drop in humidity was observed in April 2022, which could suggest dry or windy weather conditions during that time. Conversely, the higher peak was observed in May and June 2023; this could be explained by the fact that, in 2023, there was a drop in the maximum temperature in the months of May and June.

The “Cross Correlation time series” highlighted a statistically significant association between the environmental parameters examined (temperature and humidity) and the prevalence of infection. The result obtained from the correlation test between the time series of prevalence and of temperature is shown in Figure 4.

Figure 4 shows that there is a positive and significant correlation at lag −0.1667 between the time series of prevalence and that of temperature, with a correlation index equal to 0.244; this means that the effect of temperature on prevalence occurs approximately two months later.

Regarding the cross-correlation between prevalence and humidity (Figure 5), there is a significant association at lag 0.25 (CFF 0.274); this means that a decrease or increase in prevalence happens approximately 2 and a half months before a decrease or increase in humidity.

In both cases (T and U), the CCF values are positive indicating a direct proportionality between the time series.

The Spearman correlation coefficient (rho) is equal to 0.1035; this value indicates a weak positive correlation between the two variables (T and U). The result obtained indicates that the differences between the variables were not statistically significant and the null hypothesis cannot be rejected (*p*-value > 0.05).

The generalized mixed-effects regression models, built considering as covariates the humidity and temperature values at lag −2 and lag −5, respectively, have shown that both humidity and temperature are significant predictors for the prevalence of infection. In fact, it was found that with a unit increase in the humidity value, there is an average increase in prevalence of 0.06 with a delay of two months, while with regard to temperature, it was found that with a unit increase in the T value, there is an average increase in prevalence of 0.082 with a delay of 5 months (*p*-value < 0.05). The temperature and humidity values at lag −5 and −2, respectively, were chosen for the construction of the models, as they were the first values that showed satisfactory results.

Table 1 shows the relative risks of the univariate models.

The relative risks (RR) indicate that, with a unit increase in the humidity value, the risk of disease spread increases by 6.33% and that with a unit increase in the temperature value, the prevalence increases by 8.6%. The confidence interval (CI) does not contain the unit and is therefore statistically significant.

The result obtained from the univariate models was also confirmed by the construction of the multivariate model; it was evident that higher temperature and humidity values contribute positively to the spread of infection.

Table 2 shows the relative risk calculated on each covariate introduced in the multivariate model with the outcome variable represented by prevalence.

Humidity and temperature both show a positive and statistically significant effect on the dependent variable. The RR associated with humidity is approximately 1.05 indicating that, for each unit increase in humidity, the probability of disease spread increases by 4.56%. The RR associated with temperature is approximately 1.08, indicating that, for each unit increase in temperature, the probability of disease spread increases by 7.84%. These results suggest that both factors are relevant in the analyzed model and influence the probability of infection in a positive way.

Eventually, to evaluate the possible risk of proximity or otherwise to wetlands and/or bodies of water, descriptive analysis was carried out and Fisher’s test was applied (Table 3, Table 4, Table 5, Table 6, Table 7 and Table 8).

A statistically significant association was found (*p*-value < 0.05) between the outcome of the farm and the proximity to water bodies for the years 2018, 2019 and 2021, while the association with wetlands was significant in 2018, 2020 and 2021.

The ODDS RATIOs (OR), obtained from the models, have highlighted that proximity to water bodies appears to be a risk factor; specifically, Table 9 reports the summary of the ODDS RATIO with the confidence intervals and relative *p*-value.

The proximity of farms to watercourses represents a risk for the development of arboviruses infections. In particular, in 2018, 2019 and 2021 the risk of developing the disease for farms located in the 5 km buffer zone containing watercourses was found to be 2.75, 2.95 and 3.95 times higher, respectively, than for farms located outside the 5 km buffer zone containing watercourses.

## 4. Discussion

Our results highlight the significant role of climatic factors in the transmission dynamics of arboviruses: the average temperature is the most effective factor in determining the spread of this infection, as seen in previous studies conducted on other arboviruses [11]. Insects, being ectothermic, develop and mature more rapidly in warm environments. Rising temperatures reduce the time to develop from larvae to adults, resulting in an increase in the mosquito population. This leads to a greater probability of transmission of the virus. Furthermore, higher temperatures also accelerate the replication of arboviruses inside the vector. This reduces the “extrinsic incubation time, i.e., the period during which the virus becomes infectious within the vector, thus increasing the probability that the vector will infect humans and other animals. Humidity, from the results obtained, also plays a fundamental role: where humidity increases, the presence of infection is increased, especially in the hot months. High humidity, particularly after periods of heavy rainfall or in the presence of stagnant bodies of water, provides the ideal habitat for insects to breed [20,21]. Pools of water, necessary for egg hatching, increase in number with higher humidity levels, leading to higher vector densities [22]. This aspect was confirmed by our study as proximity to bodies of water represents a three times greater risk of developing the infection compared to areas far from them. Prolonged high humidity can also extend the breeding season of insects, meaning that transmission of the virus can continue for a longer period of time during the year In this regard, in Campania, in recent years, cases of vector-borne diseases have been found in the winter period, months that had been rarely affected in the past and, therefore, are not affected by the surveillance plans. It is important to underline that humidity, in combination with temperature, influences the spread of arboviruses mainly through the effect on the reproduction of vectors. In optimal conditions of humidity and temperature, there is not only an increase in insect populations but also increased sting activity. Insects tend to be more active in higher humidity conditions, thus increasing the likelihood of transmission of the virus to human and animal hosts.

The results obtained must be considered in light of the possible limitations of potential biases. The information presented in the various information systems could not be complete or entirely correct. In this regard, on the positive samples, the geographical coordinates, reported on the accompanying sheets, are always present, since at the time of confirmation, they are verified and inserted where missing, while for the negative samples, they are occasionally absent. Over time, the implementation has improved thanks to the greater degree of awareness and sensitization on the importance of data traceability.

Global warming, with rising temperatures and changes in humidity, is a key factor that is expanding the risk of arboviruses into previously unaffected areas. The magnitude and pace of these changes are expected to increase without innovative vector control or significant reduction in anthropogenic factors contributing to global warming.

The policy implications of these findings are particularly relevant for public health and vector-borne disease prevention policies. Governments must address the health emergency with proactive and climate-adaptive strategies, implementing advanced and timely monitoring systems capable of responding to the new challenges posed by the spread of vectors in non-traditional areas. Health policies must be flexible and able to adapt to seasonal and geographical changes in vector spread. It is essential that vector control policies are dynamic and oriented towards a One Health approach, integrating veterinary, human and environmental surveillance [13]. Surveillance plans must be adapted to territorial specificities, taking into account local climatic factors, vulnerable areas and ecological characteristics that influence the presence and spread of vectors.

The challenge of understanding the trajectory of Arbovirus transmission in the future requires careful consideration not only how environmental changes affect current biological systems, but also how these systems will evolve and interact with future climates [23]. To address this challenge, environmental surveillance and preventive measures such as vector control become crucial to limit the spread of Arboviruses in vulnerable areas. Understanding the expected geographical and distributional changes in disease transmission due to climate change, together with effective adaptation strategies, is essential to mitigate future impacts on Public Health. It is fundamental to give guidelines to Governments in order to modulate surveillance plans based on the specific risk assessment for each territorial scenario [24]. Given the findings obtained, we propose future research to broadly explore adaptation strategies that address these climate challenges. Such efforts would provide comprehensive insights that are crucial to developing sound public health policies. Future studies could adopt a One Health approach or leverage the UNEP framework (The global authority for the environment with programs focusing on climate, nature, pollution, sustainable development and more) to explore different responses to Arboviruses, thus enriching the dialog between science climate and public health policy [22]. Given the epidemiology of Arbovirosis and the complex human/animal/environment interactions, preventive medicine cannot represent the only effective method in managing such events. To establish common actions that respond to the challenges and objectives from a One Health perspective, a shared strategic approach is essential, integrating veterinary (animal) surveillance with human and environmental (entomological) surveillance.

## 5. Conclusions

In conclusion, climate warming and land use changes have contributed to the expansion of the geographical distribution of endemic arboviruses in Europe, increasing the risk of epidemic outbreaks among humans and animals. At the same time, increased international travel and global trade have exposed Europe to a rising risk of the introduction of exotic arthropod vectors and autochthonous transmission of arboviruses, such as dengue, chikungunya, yellow fever and Zika virus. These arboviruses, originally from tropical and subtropical regions, may now establish themselves in new areas, increasing the risk of epidemics. To improve surveillance and manage arboviruses risks, the adoption of advanced predictive models and the use of new technologies are crucial. Recently, the integration of climate and biological simulation models, such as the MaxEnt model for the analysis of potential vector distribution, has demonstrated its effectiveness in predicting areas at high risk of transmission [24]. These models, which combine climate, ecological and vector distribution data, can provide accurate risk maps that help health authorities focus on the most vulnerable areas. Furthermore, the use of innovative techniques, such as machine learning, is expanding forecasting capabilities, allowing for more precise and dynamic prediction of disease outbreaks. Models based on random forests or support vector machines are able to analyze large amounts of data from disparate sources, such as changes in human migratory habits, satellite data on climate and environmental information, to identify emerging trends and predict the appearance of new vectors or outbreaks [24]. From a diagnostic point of view, laboratory techniques are improving thanks to the use of new approaches, such as biosensors and multiplex PCR, which allow for the simultaneous detection of multiple pathogens with high sensitivity and specificity, reducing the time to diagnosis and increasing monitoring capacity. Furthermore, next-generation sequencing (NGS) is revolutionizing the study of viruses, allowing for a more precise mapping of their evolution and spread. Finally, synergy between existing monitoring systems, the use of advanced predictive models, new diagnostic technologies and increased international collaboration are essential to effectively address future threats. The combination of multidisciplinary interventions, ranging from vector control to vaccine research and rapid diagnostics, is key to ensuring a timely and sustainable response to arboviral diseases. In this context, integrated arboviruses surveillance is crucial for prevention and control. Timely monitoring of the introduction of new vectors and disease outbreaks is essential to take appropriate containment measures and reduce the risk of human transmission, especially through blood, tissue and organ donations. However, identifying emerging outbreaks remains a complex challenge, requiring a high level of awareness, advanced diagnostic capabilities and effective laboratory support, especially for pathogens that are not yet considered an immediate threat but could prove highly dangerous. In recent decades, the world has witnessed an unprecedented surge in epidemic arboviral diseases, fueled by the combination of urbanization, globalization, and international mobility. This phenomenon is a clear alarm bell for governments, research institutions, funders, and the World Health Organization (WHO), who must step up efforts to improve surveillance and research on vector-borne diseases. It is essential to develop unified and targeted responses for diagnostics, vaccines, biological targets, immune responses, environmental determinants, and vector control. Global collaboration and integration of resources are key to addressing this growing threat. Combining existing and effective interventions against different arboviral diseases is the most sustainable and efficient strategy to reduce the incidence of these infections. In this scenario, building international alliances and adopting shared strategies will be crucial to ensure rapid and effective solutions. For example, the integrated One Health approach, which combines human, animal and environmental health, has been successfully implemented by organizations such as WHO, FAO and OIE, helping to monitor and prevent zoonotic diseases such as dengue and malaria through a collaborative approach that promotes joint work between different sectors [25]. Furthermore, the WHO Global Vector Control Response has provided a global framework for vector control, encouraging the adoption of sustainable practices for vector surveillance and management, such as the use of insecticides and technologies to monitor mosquito populations, which are essential to contain the spread of diseases such as chikungunya. Finally, the Partnership for Dengue Control initiative has enabled cooperation between governments, academia and the private sector to develop innovative solutions, such as dengue vaccination and the use of genetically modified mosquitoes to reduce vector populations, demonstrating how technological innovation can be a crucial element in the control of vector-borne diseases. These examples of international alliances and integrated strategic approaches show how cooperation and resource sharing are essential to developing global and timely solutions to address threats related to climate change and vector expansion.

The implementation of integrated arboviruses surveillance programs is crucial to adopt adequate and effective control measures to avoid the introduction and/or spread of arboviruses as well as to prevent human transmission via blood, tissue and organ donations. However, the identification of emerging outbreaks is challenging and requires a high degree of awareness and laboratory capability, especially for the most neglected but potentially threatening pathogens. The last five decades have seen an unprecedented emergence in epidemic arboviral diseases, resulting from the triad of the modern world, including urbanization, globalization and international mobility [26]. This is an alert for governments, academia, funders and WHO to strengthen programs and improve research on vector-borne diseases. The common features of these diseases should stimulate similar research themes for diagnostics, vaccines, biological targets and immune responses, environmental determinants and vector control measures. Combining interventions known to be effective against multiple arboviral diseases will offer the most cost-effective and sustainable strategy to reduce these infections. Therefore, new global alliances are needed to enable the combination of efforts and resources for more effective and timely solutions.

## Figures and Tables

**Figure 1 microorganisms-13-00449-f001:**
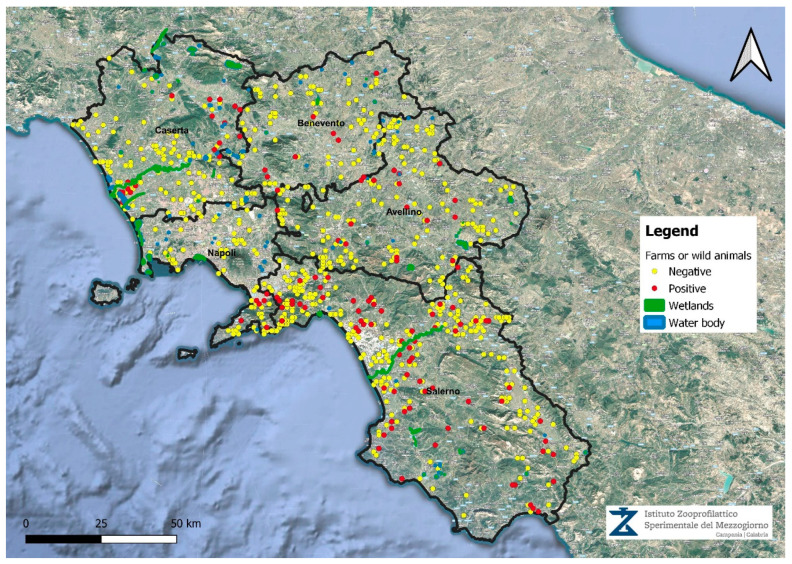
Distribution of positive cases in the period examined of the Campania Region.

**Figure 2 microorganisms-13-00449-f002:**
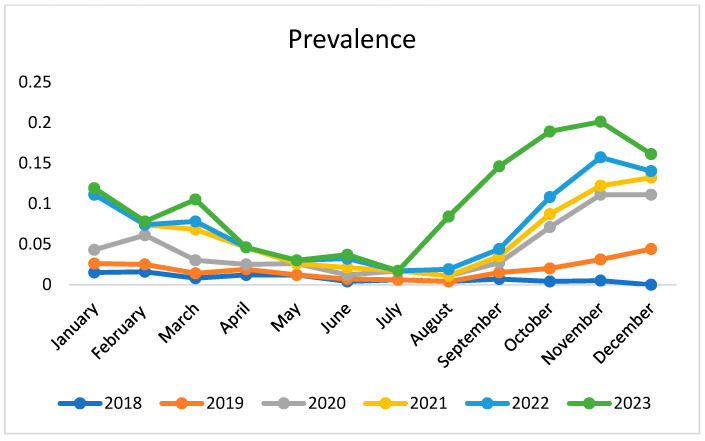
Trend of monthly prevalence in various years in the Campania region.

**Figure 3 microorganisms-13-00449-f003:**
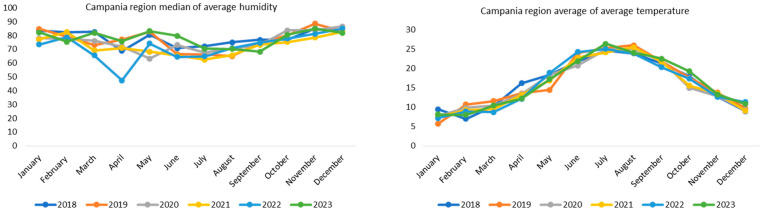
Trends in monthly humidity and temperature over the years in the Campania region.

**Figure 4 microorganisms-13-00449-f004:**
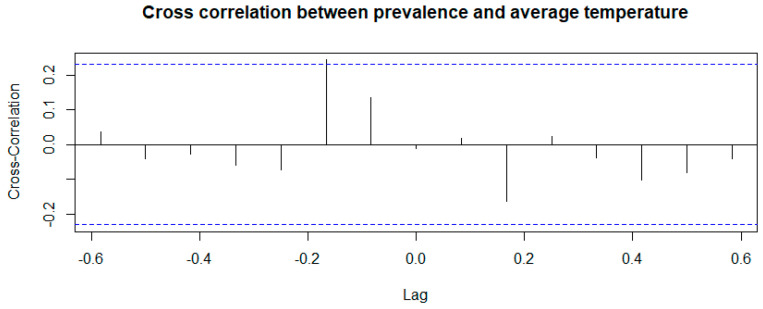
Correlation test between the time series of prevalence and of temperature.

**Figure 5 microorganisms-13-00449-f005:**
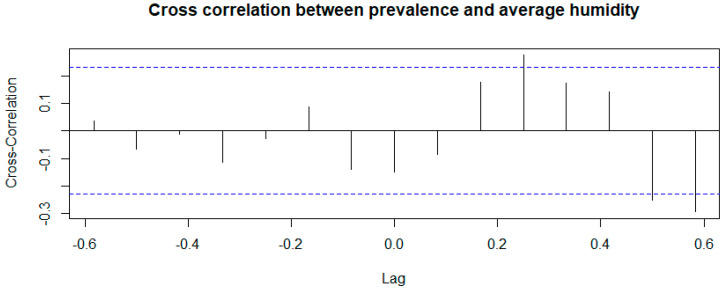
Correlation test between the time series of prevalence and of humidity.

**Table 1 microorganisms-13-00449-t001:** Relative risks of the univariate models.

Covariate	RR	Lower CI	Upper_CI	*p*-Values
Humidity	1.06335	1.028496	1.099384476	<0.05
Temperature	1.085893	1.03227392	1.142298535	<0.05

**Table 2 microorganisms-13-00449-t002:** Relative risks of the multivariate model.

Covariate	RR	Lower CI	Upper_CI	*p*-Values
Humidity	1.045641	1.023202	1.0685722066	<0.05
Temperature	1.078380	1.055080	1.1021945110	<0.05

**Table 3 microorganisms-13-00449-t003:** 2018.

	N (N = 1878)	P (N = 47)	*p*-Values
**weatland**			
no	1437 (76.5%)	46 (97.9%)	<0.05
yes	441 (23.5%)	1 (2.1%)	
**bodies of water**			
no	1362 (72.5%)	23 (48.9%)	<0.05
yes	516 (27.5%)	24 (51.1%)	

**Table 4 microorganisms-13-00449-t004:** 2019.

	N (N = 1866)	P (N = 38)	*p*-Values
**weatland**			
no	1347 (73.6%)	32 (84.2%)	0.191
yes	492 (26.4%)	6 (15.8%)	
**bodies of water**			
no	1357 (72.7%)	18 (47.4%)	<0.05
yes	509 (27.3%)	20 (52.6%)	

**Table 5 microorganisms-13-00449-t005:** 2020.

	N (N = 925)	P (N = 62)	*p*-Values
**weatland**			
no	661 (71.5%)	55 (88.7%)	<0.05
yes	264 (28.5%)	7 (11.3%)	
**bodies of water**			
no	722 (78.1%)	46 (74.2%)	0.527
yes	203 (21.9%)	16 (25.8%)	

**Table 6 microorganisms-13-00449-t006:** 2021.

	N (N = 738)	P (N = 28)	*p*-Values
**weatland**			
no	533 (72.2%)	27 (96.4%)	<0.05
yes	205 (27.8%)	1 (3.6%)	
**bodies of water**			
no	582 (78.9%)	14 (50.0%)	<0.05
yes	156 (21.1%)	14 (50.0%)	

**Table 7 microorganisms-13-00449-t007:** 2022.

	N (N = 638)	P (N = 25)	*p*-Values
**weatland**			
no	464 (72.7%)	16 (64.0%)	0.363
yes	174 (27.3%)	9 (36.0%)	
**bodies of water**			
no	467 (73.2%)	14 (56.0%)	0.0685
yes	171 (26.8%)	11 (44.0%)	

**Table 8 microorganisms-13-00449-t008:** 2023.

	N (N = 680)	P (N = 65)	*p*-Values
**weatland**			
no	488 (71.8%)	51 (78.5%)	0.31
yes	192 (28.2%)	14 (21.5%)	
**bodies of water**			
no	494 (72.6%)	50 (76.9%)	0.559
yes	186 (27.4%)	15 (23.1%)	

**Table 9 microorganisms-13-00449-t009:** OR = Odds Ratio, IC = Confidence Interval 95%. * = iteration between the two variables.

Covariate	Year *	OR	IC 95%	*p*-Value
proximity to bodies of water	2018	2.75	1.53–4.94	<0.05
proximity to bodies of water	2019	2.95	1.54–5.76	<0.05
proximity to bodies of water	2020	1.23	0.66–2.18	>0.05
proximity to bodies of water	2021	3.62	1.68–7.83	<0.05
proximity to bodies of water	2022	2.14	0.93–4.80	>0.05
proximity to bodies of water	2023	0.79	0.42–1.41	>0.05

## Data Availability

The original contributions presented in this study are included in the article. Further inquiries can be directed to the corresponding author.

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
