# Peer review of "Temporal Analysis of Climate Change Impact on the Spread and Prevalence of Vector-Borne Diseases in Campania (2018–2023)"

_microorganisms, 2025, doi:10.3390/microorganisms13020449_

Round 1
Reviewer 1 Report
Comments and Suggestions for Authors
The work is well structured and the quality of english is good. Also the topic, very interesting, is correctly reported and showed to the reader. Anyway some little changes have to be performed in order to go ongoing with the publication. If Authors will follow the suggestion given, I will certainly recommand the paper for the publication.
Introduction. In this section, Authors give a brief over view of the topic before speaking about the situation in Italy. I suggest you to improve the first part, giving a better overview of the problem all over the world. In fact, the reader in this way not have a complete comprehension of the importance of the vector-borne diseases in the other countries and may think that are concentrated in just some regions of the Mediaterrean Area.
At the same time, it could be interesting to explain how the monitoring of vector borne is possible using new technologies such as GIS and Remote Sensing. These interesting tolls use satellite data and their interpretation for predict habitat conditions and, in this way, the possible presence of vectors (suitability maps; please see here: https://doi.org/10.3389/fvets.2024.1383320; https://veterinariaitaliana.izs.it/index.php/VetIt/article/view/3481;
- https://doi.org/10.1371/journal.pone.0219072. This topic is strictly re
- lated with the One Health concept because the prediction of suitability areas may help government institution to organize focused controls and preventive misures for human health conservation (Public sector can have great information to analyze Public Health diseases).
Material and methods. I advice you to insert a figure about the study area, the same as you reported in the results section without the results is sufficient.
Line 133. When you speak about SIGLA software, you refer to the software used by the Experimental Zooprophylactic Istitute in Italy? if yes, please specified it.
Conclusion section. I think that the core message of the paragraph is not so effectively emphasized, highlighting the critical role of surveillance, research, and collaboration. Please reformulate the first part of the paragraph. In particular, the sentences are now better connected, creating a smoother and more logical flow of ideas.
Line 396-398. These considerations may be more critical analysed reporting pratical examples that may help future investigations. New softwares? New models? New laboratory techniques?
Author Response
The work is well structured and the quality of english is good. Also the topic, very interesting, is correctly reported and showed to the reader. Anyway some little changes have to be performed in order to go ongoing with the publication. If Authors will follow the suggestion given, I will certainly recommand the paper for the publication.
Introduction. In this section, Authors give a brief over view of the topic before speaking about the situation in Italy. I suggest you to improve the first part, giving a better overview of the problem all over the world. In fact, the reader in this way not have a complete comprehension of the importance of the vector-borne diseases in the other countries and may think that are concentrated in just some regions of the Mediaterrean Area.
Answer: php/VetIt/article/view/3481;
mo inserire questa nel manoscritto2024 e 2025o precipitazioni, direzione del vento, bagnatura f
Climate change has a direct impact on the distribution and activity of vectors, influencing factors such as temperature, precipitation and humidity, which are crucial for the life cycle of insects and their ability to transmit pathogens. Rising global temperatures, more extreme weather events and changing precipitation patterns are factors that are modifying the distribution of infectious diseases worldwide (Ryan et al., 2019; Estrada-Peña et al., 2020). Regions at risk include not only the tropics, but also temperate and subtropical areas, which could see an expansion of vectors and a consequent increase in vector-borne disease cases. Recent studies have documented the emergence of previously geographically restricted diseases, with malaria and dengue-carrying mosquitoes now found in territories further north, such as Europe and the United States (Patz et al., 2005; Khormi et al., 2014). For example, the spread of dengue in areas such as the Mediterranean, which were not previously subject to this disease, represents clear evidence of the influence of global warming on vector-borne diseases (Semenza et al., 2018). Furthermore, the impact of climate change is not limited to the simple geographical expansion of vectors, but can also alter transmission patterns, seasonality and intensity of epidemics. Longer heat periods and heavy rainfall can extend the vector activity season, increasing the likelihood of infectious disease transmission. The situation is further complicated by globalization, which facilitates the spread of diseases through international travel and trade (Tatem et al., 2006). In this context, a global view of vector-borne diseases is essential, as climate change is creating new risks not only for tropical regions, but also for countries with temperate climates. An analysis of global trends is therefore essential to fully understand the implications of climate change on public health.
At the same time, it could be interesting to explain how the monitoring of vector borne is possible using new technologies such as GIS and Remote Sensing. These interesting tolls use satellite data and their interpretation for predict habitat conditions and, in this way, the possible presence of vectors (suitability maps; please see here:
https://doi.org/10.3389/fvets.2024.1383320; https://veterinariaitaliana.izs.it/index.php/VetIt/article/view/3481; https://doi.org/10.1371/journal.pone.0219072. This topic is strictly related with the One Health concept because the prediction of suitability areas may help government institution to organize focused controls and preventive misures for human health conservation (Public sector can have great information to analyze Public Health diseases).
A promising approach to address this challenge is the use of advanced technologies, such as Geographic Information Systems (GIS) and Remote Sensing, which allow monitoring and forecasting the distribution of vectors in relation to ecological and climatic variables. The integration of satellite data with environmental suitability models allows mapping areas at risk of disease transmission, providing crucial information for planning and im-plementing targeted control and prevention measures. The predictive approach, which exploits these technological tools, not only optimizes the management of health resources, but is part of the concept of "One Health", which recognizes the connection between hu-man, animal and environmental health. A concrete example of the application of these technologies is represented by the study by Ippoliti et al. (2024), which uses a multivariate clustering method to define ecological regions in Italy and improve the surveillance of vector-borne diseases. This study demonstrates how predictive analytics can support health policies, improving the capacity to respond to emerging diseases and reducing ri-sks to public health.Ippoliti et al.
Material and methods. I advice you to insert a figure about the study area, the same as you reported in the results section without the results is sufficient.
We did not understand the request
Line 133. When you speak about SIGLA software, you refer to the software used by the Experimental Zooprophylactic Istitute in Italy? if yes, please specified it.
YES it is specified on line 118 if you want we can also report it on line 133
Conclusion section. I think that the core message of the paragraph is not so effectively emphasized, highlighting the critical role of surveillance, research, and collaboration. Please reformulate the first part of the paragraph. In particular, the sentences are now better connected, creating a smoother and more logical flow of ideas
In conclusion, Climate warming and land use changes have contributed to the expansion of the geographical distribution of endemic arboviruses in Europe, increasing the risk of epidemic outbreaks among humans and animals. At the same time, increased internatio-nal travel and global trade have exposed Europe to an increasing risk of introduction of exotic arthropod vectors and autochthonous transmission of arboviruses, such as dengue, chikungunya, yellow fever and Zika virus. These arboviruses, originally from tropical and subtropical regions, may now establish themselves in new areas, increasing the risk of epidemics. To improve surveillance and management of arbovirus risks, the adoption of advanced predictive models and the use of new technologies are crucial. Recently, the in-tegration of climate and biological simulation models, such as the MaxEnt model for the analysis of potential vector distribution, has demonstrated its effectiveness in predicting areas at high risk of transmission (Elith et al., 2011). These models, which combine clima-te, ecological and vector distribution data, can provide accurate risk maps that help health authorities focus on the most vulnerable areas. Furthermore, the use of innovative techni-ques such as machine learning is expanding forecasting capabilities, allowing for more precise and dynamic prediction of disease outbreaks. Models based on random forests or support vector machines are able to analyze large amounts of data from disparate sources, such as changes in human migratory habits, satellite data on climate and environmental information, to identify emerging trends and predict the appearance of new vectors or outbreaks (Chadee et al., 2019). From a diagnostic point of view, laboratory techniques are improving thanks to the use of new approaches such as biosensors and multiplex PCR, which allow for the simultaneous detection of multiple pathogens with high sensitivity and specificity, reducing the time to diagnosis and increasing monitoring capacity. Furthermore, next-generation sequencing (NGS) is revolutionizing the study of viruses, allowing for a more precise mapping of their evolution and spread. Finally, sy-nergy between existing monitoring systems, the use of advanced predictive models, new diagnostic technologies and increased international collaboration are essential to effecti-vely address future threats. The combination of multidisciplinary interventions, ranging from vector control to vaccine research and rapid diagnostics, is key to ensuring a timely and sustainable response to arboviral diseases. In this context, integrated arbovirus sur-veillance is crucial for prevention and control. Timely monitoring of the introduction of new vectors and disease outbreaks is essential to take appropriate containment measures and reduce the risk of human transmission, especially through blood, tissue and organ donations. However, identifying emerging outbreaks remains a complex challenge, requi-ring a high level of awareness, advanced diagnostic capabilities and effective laboratory support, especially for pathogens that are not yet considered an immediate threat but could prove highly dangerous. In recent decades, the world has witnessed an unprece-dented surge in epidemic arboviral diseases, fueled by the combination of urbanization, globalization, and international mobility. This phenomenon is a clear alarm bell for go-vernments, research institutions, funders, and the World Health Organization (WHO), who must step up efforts to improve surveillance and research on vector-borne diseases. It is essential to develop unified and targeted responses for diagnostics, vaccines, biological targets, immune responses, environmental determinants, and vector control.
Line 396-398. These considerations may be more critical analysed reporting pratical examples that may help future investigations. New softwares? New models? New laboratory techniques? Ok done
Recently, the integration of climate and biological simulation models, such as the MaxEnt model for the analysis of potential vector distribution, has demonstrated its effectiveness in predicting areas at high risk of transmission (Elith et al., 2011). These models, which combine climate, ecological and vector distribution data, can provide accurate risk maps that help health authorities focus on the most vulnerable areas. Furthermore, the use of innovative techniques such as machine learning is expanding forecasting capabilities, allowing for more precise and dynamic prediction of disease outbreaks. Models based on random forests or support vector machines are able to analyze large amounts of data from disparate sources, such as changes in human migratory habits, satellite data on climate and environmental information, to identify emerging trends and predict the appearance of new vectors or outbreaks (Chadee et al., 2019). From a diagnostic point of view, laboratory techniques are improving thanks to the use of new approaches such as biosensors and multiplex PCR, which allow for the simultaneous detection of multiple pathogens with high sensitivity and specificity, reducing the time to diagnosis and increasing monitoring capacity. Furthermore, next-generation sequencing (NGS) is revolutionizing the study of viruses, allowing for a more precise mapping of their evolution and spread.
Reviewer 2 Report
Comments and Suggestions for Authors
Abstract:
The phrase "to provide useful elements to the Competent Authorities" can be improved to "to offer actionable insights for health authorities."
Consider mentioning the key findings quantitatively in the abstract, such as the specific percentage increase in disease prevalence related to temperature and humidity changes.
Introduction
The reference to climate change influencing the spread of arboviruses is appropriate, but the sentence "Many indeed, in fact, which were once considered exotic..." should be rephrased for clarity.
The objectives are clearly stated, but it might be helpful to explicitly mention the hypotheses being tested.
Results
The discussion of the correlation between temperature, humidity, and prevalence is clear. However, explicitly state whether any other environmental factors were tested but found insignificant.
Discussion:
The statement "Global warming, with rising temperatures and changes in humidity, is a key factor..." could be supported by additional references for robustness.
Consider expanding on the potential policy implications of your findings, particularly regarding surveillance and control measures.
Conclusion:
The mention of "new global alliances" is forward-thinking but would benefit from a brief example or reference to existing successful alliances.
Author Response
Abstract:
The phrase "to provide useful elements to the Competent Authorities" can be improved to "to offer actionable insights for health authorities." Done
Consider mentioning the key findings quantitatively in the abstract, such as the specific percentage increase in disease prevalence related to temperature and humidity changes.
Done “The generalized linear mixed model showed that for each unit increase in humidity and temperature, the probability of disease spread increased by 4.56% and 7.84%, respectively”
Introduction
The reference to climate change influencing the spread of arboviruses is appropriate, but the sentence "Many indeed, in fact, which were once considered exotic..." should be rephrased for clarity.…
Answer
The intruduction is changed:
Vector-borne diseases constitute an important public health problem: the WHO estimates that every year they cause over a billion human cases and 1 million deaths, representing approximately 17% of total cases of communicable diseases [1]. There are over 100 viruses classified as arboviruses capable of causing human disease. Climate change has a direct impact on the distribution and activity of vectors, influencing factors such as temperature, precipitation and humidity, which are crucial for the life cycle of insects and their ability to transmit pathogens. Rising global temperatures, more extreme weather events and changing precipitation patterns are factors that are modifying the distribution of infec-tious diseases worldwide (Ryan et al., 2019; Estrada-Peña et al., 2020). Regions at risk in-clude not only the tropics, but also temperate and subtropical areas, which could see an expansion of vectors and a consequent increase in vector-borne disease cases. Recent stu-dies have documented the emergence of previously geographically restricted diseases, with malaria and dengue-carrying mosquitoes now found in territories further north, such as Europe and the United States (Patz et al., 2005; Khormi, et al., 2014). For exam-ple, the spread of dengue in areas such as the Mediterranean, which were not previously subject to this disease, represents clear evidence of the influence of global warming on vector-borne diseases (Semenza et al., 2018). Furthermore, the impact of climate change is not limited to the simple geographical expansion of vectors, but can also alter transmis-sion patterns, seasonality and intensity of epidemics. Longer heat periods and heavy rainfall can extend the vector activity season, increasing the likelihood of infectious disea-se transmission. The situation is further complicated by globalization, which facilitates the spread of diseases through international travel and trade (Tatem et al., 2006). In this context, a global view of vector-borne diseases is essential, as climate change is creating new risks not only for tropical regions, but also for countries with temperate climates. An analysis of global trends is therefore essential to fully understand the implications of cli-mate change on public health. A promising approach to address this challenge is the use of advanced technologies, such as Geographic Information Systems (GIS) and Remote Sensing, which allow monitoring and forecasting the distribution of vectors in relation to ecological and climatic variables. The integration of satellite data with environmental suitability models allows mapping areas at risk of disease transmission, providing crucial information for planning and implementing targeted control and prevention measures. The predictive approach, which exploits these technological tools, not only optimizes the management of health resources, but is part of the concept of "One Health", which reco-gnizes the connection between human, animal and environmental health. A concrete example of the application of these technologies is represented by the study by Ippoliti et al. (2024), which uses a multivariate clustering method to define ecological regions in Italy and improve the surveillance of vector-borne diseases. This study demonstrates how pre-dictive analytics can support health policies, improving the capacity to respond to emer-ging diseases and reducing risks to public health. In Italy, there are both native arboviru-ses, including West Nile disease, Usutu virus infection, Toscana virus infection and tick-borne viral encephalitis, and predominantly imported arboviruses, such as the infec-tions caused from Chikungunya, Dengue and Zika viruses, Blue Tongue (BTV) [2]. Usutu (USUV) and West Nile (WNV) viruses are mosquito-borne neurotropic flaviviruses belon-ging to the Flaviviridae family and the Flavivirus genus. The natural transmission cycle of WNV and USUV involves mosquitoes and birds, while mammals are believed to be acci-dental hosts. WNV is now the most widespread virus among the flaviviruses reported on almost all continents. In the last two decades their presence has been confirmed in several European and Mediterranean countries, becoming endemic in some of them. USUV was originally isolated in South Africa and first detected in Europe in 2001, where it caused severe mortality in Austrian blackbird populations. In the following years, USUV spread to central and eastern Europe, affecting wild birds, including populations from Italy, where the virus has probably been circulating since 1996. The USUV was also first detec-ted in the UK in 2020 [3]. Bluetongue is an arthropod-borne viral infection that is notifiable in several countries and causes significant economic losses and serious concerns for the ruminant trade. BTV is a non-contagious RNA virus spread by Culicoides bite, which has significantly invaded Europe in recent decades [4]. BTV was responsible for well-known epidemics in the late 1990s and early 2000s and is now considered an endemic and re-emerging virus throughout Europe. Due to the severe symptoms exhibited by infected animals, BTV infection causes economic losses and impacts international livestock trade (estimated at up to $3 billion) [5]. Although BTV primarily results in clinical symptoms in sheep, cattle serve as reservoir hosts and play an important role in the transmission and epidemiology of BTV because they exhibit prolonged viremia (rarely associated with re-productive failure and low milk production) [5]. Surveillance for this virus relies on iden-tification of the pathogen in vectors and symptomatic hosts, as well as the detection of specific antibodies. There are numerous factors that influence the transmission and di-stribution of arboviruses, linked to the dynamics and interactions among pathogen, vec-tor, host and environment; climatic conditions have direct and indirect influences on the competence of the vector (the ability to acquire, maintain and transmit the virus), on the dynamics of the vector population and on the rate of replication of the virus within the mosquito. The dynamic evolution of ecosystems together with climate changes have played and still play an important role in the spread of these diseases, contributing to their spread in countries where these pathologies were not present and influencing their persistence in new areas. The impact of climatic factors (temperature, precipitation, rela-tive humidity and winds) on the epidemiology of arboviruses, in fact, is increasing consi-derably as a function of the current continuous climate change which influences the emergence of arboviruses such as malaria, dengue and WNV by altering its frequency, ex-tent, distribution and seasonality. Many indeed, in fact, which were once considered exo-tic in our country, have now become epidemic and/or sometimes endemic. The Intergo-vernmental Panel on Climate Change (IPCC) lists vector-borne diseases among the events most likely to change due to global warming. On the other hand, these diseases, particu-larly sensitive to climate fluctuations, could act as an alarm to focus attention on the threats of climate change [6]. In this regard, Governments are obliged to consider the effect of climate change in their prevention, control and surveillance perspectives in all coun-tries of the world. To mitigate the impacts of CC on the transmission ecology of vector di-seases, it is important to implement effective surveillance and control measures that can-not ignore the study of the effect of climate change on the onset and spread of these infec-tions; in order to modulate and intensify early detection in certain periods of the year, thus ensuring efficient and effective use of the resources available to protect health.
The “One Health” approach, based on the cooperation and interdisciplinarity of various professional figures, could be the only valid and truly effective alternative to deal with possible threats emerging from the human-animal-environment interface [7]. The organi-zation of a territorial cooperation network constitutes a valid innovative reference model in the integration of the different activities and experiences of human medicine and vete-rinary medicine through the intensification and integration of animal surveillance and prevention plans with human and environmental (entomological). In recent decades, an increase in cases of vector-borne infections, both emerging and non-emerging, has been observed in many Italian regions [7]. In the Campania region, there has been an increase in WND cases in recent years versus no, or almost no, detection of infection in the years prior to 2022. The objective of this study is to determine whether climatic parameters, such as temperature and humidity, representing risk factors for the spread of Arbovirosis in the Campania region in the years 2018-2023. The results may be useful to the Competent Au-thorities to modulate existing surveillance plans.
The objectives are clearly stated, but it might be helpful to explicitly mention the hypotheses being tested.
Answer
Below is the new wording: we ask if it is okay and if we should insert this into the manuscript.
The aim of this study is to determine whether climate parameters, such as temperature and humidity, represent risk factors for the spread of Arbovirosis in the Campania region in the years 2018-2023. In this regard, the possible association between humidity (U) and Temperature (T) and the outcome of the study (prevalence of arbovirosis P) was evaluated. The two hypotheses were then tested (H0: T and U not correlated to P; H1: T and U correlated to P. The risk linked to these environmental parameters taken into consideration was then estimated. The results may be useful to the Competent Authorities for existing surveillance modules
Results
The discussion of the correlation between temperature, humidity, and prevalence is clear. However, explicitly state whether any other environmental factors were tested but found insignificant.
Answer
For this work, no other factors related to climate were taken into consideration. We hope in the future to be able to examine other factors such as precipitation, wind direction, leaf wetness, expanding the dataset both in the temporal range and on the arbovirosis themselves.
Discussion:
The statement "Global warming, with rising temperatures and changes in humidity, is a key factor..." could be supported by additional references for robustness.
Consider expanding on the potential policy implications of your findings, particularly regarding surveillance and control measures.
Answer
We have inserted this paragraph:
The policy implications of these findings are particularly relevant for public health and vector-borne disease prevention policies. Governments must address the health emergency with proactive and climate-adaptive strategies, implementing advanced and timely monitoring systems, capable of responding to the new challenges posed by the spread of vectors in non-traditional areas. Health policies must be flexible and able to adapt to seasonal and geographical changes in vector spread. It is essential that vector control policies are dynamic and oriented towards a One Health approach, integrating veterinary, human and environmental surveillance [6]. Surveillance plans must be adapted to territorial specificities, taking into account local climatic factors, vulnerable areas and ecological characteristics that influence the presence and spread of vectors
Conclusion:
The mention of "new global alliances" is forward-thinking but would benefit from a brief example or reference to existing successful alliances. Done
Answer
The combination of multidisciplinary interventions, ranging from vector control to vacci-ne research and rapid diagnostics, is key to ensuring a timely and sustainable response to arboviral diseases. In this context, integrated arbovirus surveillance is crucial for preven-tion and control. Timely monitoring of the introduction of new vectors and disease out-breaks is essential to take appropriate containment measures and reduce the risk of hu-man transmission, especially through blood, tissue and organ donations. However, iden-tifying emerging outbreaks remains a complex challenge, requiring a high level of aware-ness, advanced diagnostic capabilities and effective laboratory support, especially for pa-thogens that are not yet considered an immediate threat but could prove highly dange-rous. In recent decades, the world has witnessed an unprecedented surge in epidemic ar-boviral diseases, fueled by the combination of urbanization, globalization, and interna-tional mobility. This phenomenon is a clear alarm bell for governments, research institu-tions, funders, and the World Health Organization (WHO), who must step up efforts to improve surveillance and research on vector-borne diseases. It is essential to develop uni-fied and targeted responses for diagnostics, vaccines, biological targets, immune respon-ses, environmental determinants, and vector control. Global collaboration and integration of resources are key to addressing this growing threat. Combining existing and effective interventions against different arboviral diseases is the most sustainable and efficient strategy to reduce the incidence of these infections. In this scenario, building international alliances and adopting shared strategies will be crucial to ensure rapid and effective solu-tions. For example, the integrated One Health approach, which combines human, animal and environmental health, has been successfully implemented by organizations such as WHO, FAO and OIE, helping to monitor and prevent zoonotic diseases such as dengue and malaria through a collaborative approach that promotes joint work between different sectors. Furthermore, the WHO Global Vector Control Response has provided a global framework for vector control, encouraging the adoption of sustainable practices for vector surveillance and management, such as the use of insecticides and technologies to monitor mosquito populations, which are essential to contain the spread of diseases such as chi-kungunya. Finally, the Partnership for Dengue Control initiative has enabled cooperation between governments, academia and the private sector to develop innovative solutions such as dengue vaccination and the use of genetically modified mosquitoes to reduce vec-tor populations, demonstrating how technological innovation can be a crucial element in the control of vector-borne diseases. These examples of international alliances and inte-grated strategic approaches show how cooperation and resource sharing are essential to developing global and timely solutions to address threats related to climate change and vector expansion.
Round 2
Reviewer 2 Report
Comments and Suggestions for Authors
Most of the points I pointed out have been corrected. Thanks for your hard work.
